# Tick Immunobiology and Extracellular Traps: An Integrative Vision to Control of Vectors

**DOI:** 10.3390/pathogens10111511

**Published:** 2021-11-19

**Authors:** Hugo Aguilar-Díaz, Rosa Estela Quiroz-Castañeda, Karina Salazar-Morales, Raquel Cossío-Bayúgar, Estefan Miranda-Miranda

**Affiliations:** 1Unidad de Artropodología, Centro Nacional de Investigación Disciplinaria en Salud Animal e Inocuidad INIFAP, Jiutepec 62574, Mexico; cossio.raquel@inifap.gob.mx (R.C.-B.); miranda.estefhan@inifap.gob.mx (E.M.-M.); 2Unidad de Anaplasmosis, Centro Nacional de Investigación Disciplinaria en Salud Animal e Inocuidad INIFAP, Jiutepec 62574, Mexico; requiroz79@yahoo.com.mx; 3Centro de Investigaciones Sobre Enfermedades Infecciosas, Instituto Nacional de Salud Pública, Cuernavaca 62100, Mexico; karina.salazar@insp.edu.mx

**Keywords:** anti-tick vaccines, tick immune system, *Rhipicephalus microplus*, extracellular traps, Extracellular Traps Formation (ETosis), Neutrophils Extracellular Traps Formation (NETosis), cattle diseases

## Abstract

Ticks are hematophagous ectoparasites that infest a diverse number of vertebrate hosts. The tick immunobiology plays a significant role in establishing and transmitting many pathogens to their hosts. To control tick infestations, the acaricide application is a commonly used method with severe environmental consequences and the selection of tick-resistant populations. With these drawbacks, new tick control methods need to be developed, and the immune system of ticks contains a plethora of potential candidates for vaccine design. Additionally, tick immunity is based on an orchestrated action of humoral and cellular immune responses. Therefore, the actors of these responses are the object of our study in this review since they are new targets in anti-tick vaccine design. We present their role in the immune response that positions them as feasible targets that can be blocked, inhibited, interfered with, and overexpressed, and then elucidate a new method to control tick infestations through the development of vaccines. We also propose Extracellular Traps Formation (ETosis) in ticks as a process to eliminate their natural enemies and those pathogens they transmit (vectorial capacity), which results attractive since they are a source of acting molecules with potential use as vaccines.

## 1. Introduction

Since recent years of the 20th century, the growing population in the world has demanded higher amounts of food. In this regard, beef cattle are a significant source of meat, constantly threatened by diseases transmitted by vectors. Ticks and tick-borne diseases (TBDs) are considered essential topics in the veterinary field [1]. They limit livestock activities in tropical and subtropical regions affecting more than 80% of cattle around the world and generating an economic impact of 14–19 billion USD/year [2]. Among arthropods, tick *Rhipicephalus microplus* is the ectoparasite hematophagous with the highest prevalence in cattle, associated with production and economic losses due to animal damages (skin damage, decreased productivity, lower working efficiency, slow growth of animals, and cost for control measures) [3].

Direct damages caused by tick bites include blood loss through engorgement, stress, anemia, inflammation, depression of immune function, and allergies caused by antigens and coagulants of the tick saliva [4]. Additionally, ticks’ saliva contains several protein families that induce toxicoses, including paralysis, possibly induced by blocking ion channels that regulate the cell membrane potential in tissues [5].

Ticks are efficient vectors and reservoirs of pathogens, including viruses, bacteria, and protozoans, transmitted to vertebrate hosts [6]. Over time, ticks and ticks-transmitted parasites have co-evolved with their hosts as part of the ecological niche’s equilibrium [7].

On the other hand, the indiscriminate use of chemical acaricides has allowed the selection of resistant populations of ticks and constant environmental pollution [8]. Since the first reports of resistance development of *Rhipicephalus* spp. to chemical compounds, this ability of ticks has evolved, and now they are resistant to almost all the available acaricides [9,10]. In this regard, the search for new tick control measures is a priority to cope with the adverse impacts of ticks and TBDs on cattle.

So far, the information about the interaction pathogen-vector and the plasticity of ticks’ immune system in response to pathogens is scarce. This review presents a general view of ticks’ immune system and the main molecules that participate in its defense. Here, we propose that discovering new mechanisms in ticks, such as the production of extracellular traps (ETosis), also provides alternatives to identify new targets to develop strategies to control ticks and TBDs. Additionally, the study of the tick immune system components demands a new and more integrative vision of the molecules that participates in a basic but evolutionarily well-adapted process.

## 2. Physical Barriers and Tissues

When a tick acquires a pathogen from an infected host, it must overcome physical (cuticle and peritrophic membrane (PM)) and tissue barriers (midgut, hemocoel, and salivary glands) and resist the tick’s innate immune responses to be successfully transmitted to another host [6,11].

The physical barriers prevent the internalization and spread of pathogens inside the tick. The cuticle is the first barrier to protect from hostile external environmental conditions, formed by a porous network of chitin polymers, along with accessory proteins, glycoproteins, and glycans [12,13]. When abrasions or physical damage occurs, the cuticle contains repair mechanisms to avoid pathogens’ internalization [12]. The PM is a selective physical barrier that transports metabolites and small molecules between the lumen and surrounding midgut tissues. In blood-sucking arthropods, such as ticks, the PM has a vital role in the persistence of selected pathogens, especially those that colonize or traverse the gut epithelial tissue [14].

On the other hand, the midgut is the first tissue of contact between blood components and ingested microorganisms, including pathogens. The function of the proteins identified in the midgut varies from nutrient transportation, anti-coagulation, detoxification, and lipids metabolism [15]. In the midgut, some proteins from the host blood may be used to produce similar molecules of tick’s immune system, as hemoglobin, which is fragmented to form small molecules with antimicrobial activity, or complement system proteins to regulate the tick gut microbiota populations [6].

Tick salivary glands and saliva components (cement precursors, enzymes, and inhibitors, histamine agonists and antagonists, prostaglandins, inhibitors of blood coagulation and platelet aggregation, and vasodilatory and immunomodulating factors) have an essential role in homeostasis regulation, inflammation, and immune response at the tick-host interface [16]. Some of these components are involved in the tick immune response, as shown in the sialomes studies, revealing the participation of antimicrobial peptides (AMPs) during the infection process of *Anaplasma phagocytophilum* in *Ixodes scapularis* [17].

## 3. The Tick Immunobiology

The first response line against pathogens in vertebrates is innate immunity, nonspecific, non-anticipatory, non-clonal, and germline-encoded. On the other hand, adaptive immunity is specific clonal, anticipatory, somatic, and creates immunological memory [18].

In ticks, the innate immune response comprises two major components: cellular defenses, which includes phagocytosis and encapsulation, and humoral responses, involving the AMPs, defensins, enzymes, among other molecules, expressed by the hemocytes, fat body, and the midgut [19]. In contrast, the adaptive immune response in ticks is controversial and is not yet elucidated. Similarly, several essential mechanisms of tick immune response to identify new molecules or targets with therapeutical applications remain unclear.

### 3.1. Tick Signaling Pathways in Immune Response

The reported tick genomic information has contributed to the study and elucidation of the immune response signaling pathways [20,21,22]. In *R. microplus*, in silico analysis revealed that the Toll, Immune deficiency (IMD), Jun-N-terminal kinase (JNK), Janus kinase/signal transducer, and activator of transcription (JAK/STAT) are the main signaling pathways that regulate the innate immune response. Several components of these pathways are homologous in other arthropods; however, many were not found in the ticks’ databases [23].

#### 3.1.1. Toll Pathway

The Toll pathway is mainly responsible for recognizing and eliminating fungi and Gram-positive bacteria in *Drosophila,* and viruses in the mosquito *Aedes aegypti,* in both cases by synthesizing AMPs secreted into the hemolymph [24,25]. In ticks, how the Toll pathway operates is unclear. Nevertheless, most Toll pathway components have been identified by genomic analyses, including the NF-kB Dorsal, which suggests the existence of a conserved mechanism in arthropods. In contrast, the DIF (NF-kB transcription factor dorsal-related immunity factor) is the only Toll pathway component not yet reported in tick species [19]. Some reports in ticks have shown up-regulation of the Toll pathway genes in the presence of heat-killed *Saccharomyces cerevisiae, Enterobacter cloacae, Micrococcus luteus,* and live *Rickettsia rickettsii.* Additionally, *Anaplasma marginale* down-regulated the gene expression, suggesting that pathogen can regulate this pathway to establish in a tick [23].

#### 3.1.2. Immune Deficiency (IMD) Pathway

In *Drosophila*, the IMD pathway activated by diaminopimelic acid (DAP)-type peptidoglycan present in the cell wall of most Gram-negative bacteria and some Gram-positive bacteria triggers the synthesis of specific AMPs [26]. In ticks, genomic analyses showed the lack of orthologs of crucial elements in this pathway as death-related ced-3/Nedd-2-like protein (DREDD), Fas-associated protein with death domain (FADD), and Transmembrane PGRP. It is essential to highlight that the losses of IMD pathway components are not exclusive to ticks since they are also absent in hemipterans and other arachnids [27]. Additionally, NF-kB/Relish and JNK pathway and some associated molecules to this process are conserved [28]. *Drosophila* suggested that many components of this pathway could participate in the immune response to the microbial challenge, stress, and epithelial injury, such as the transcriptional factor basket, encoded by the single JNK gene (bsk) [29].

On the other hand, Shaw et al. [30] showed that lipids such as POPG and PODAG (1-palmitoyl-2-oleoyl-snglycerol-3-phosphoglycerol and 1-palmitoyl-2-oleoyl diacylglycerol, respectively) derived from *Borrelia burgdorferi, A. phagocytophilum*, and *A. marginale* act as recognition of pathogen-associated patterns (PAMPs) and controls the IMD pathway activation and induces the colonization in *I. scapularis* and *Dermacentor andersoni*.

#### 3.1.3. Janus Kinase/Signal Transducer and Activator of Transcription (JAK/STAT) Pathway

The JAK/STAT pathway is well conserved in *Drosophila* and ticks. The fruit fly indirectly controls bacterial and fungal infections and is especially sensitive to viral infections [27,31]. In ticks, this pathway is activated by bacteria or protozoan pathogens and can positively or negatively regulate the infection in ticks. In *I. scapularis,* the JAK/STAT pathway has a vital role in the signaling between the gut microbiota and pathogen colonization [17,32]. Additionally, in this tick, AMPs expression in the salivary glands and hemocytes is regulated by the JAK/STAT pathway, where 5.3-kDa family members participate during the infection of *A. phagocytophilum* [6]. Negative regulation of AMPs expression, specifically in the salivary glands (ixodidin and lysozyme) and the gut and salivary glands (defensin), occurs in *R. microplus* STAT-deficient [33].

#### 3.1.4. RNA Interference (RNAi) Pathway

In *Drosophila*, there are four RNAi pathways reported, three endogenous: microRNA (miRNA), small interfering RNA (endo-siRNA), and piwi-interacting RNA (piRNA), and one exogenous small interference (siRNA) [34]. Interestingly, in *I. scapularis,* RNAi functions efficiently in many tissues. Omics-studies-based analyses revealed the presence of all components for the endogenous and exogenous RNAi machinery, including Dicer (Dcr), Argonaute (Ago), dsRNA binding proteins, exonucleases, and even RNA-dependent RNA polymerases [35]. In general terms, the RNAi mechanism consists of the enzymatic-complex activation after a viral infection. A long viral dsRNA is recognized and cleaved by Dcr to produce 21 nt siRNAs, known as viRNAs. These viRNAs transfer to Ago, which couples to other members of the RNA-induced silencing complex (RISC). Finally, only one strand of the viRNA remains coupled to RISC and guides the degradation of complementary viral RNA [36,37]. Undoubtedly, the tick RNAi system is still a new field awaiting further investigation, not only for its role in defense against infections but also for its potential in studying tick gene function and the screening and characterization of tick protective antigens [19].

On the other hand, RNAi as a molecular tool has helped elucidate the function of tick proteins and identify novel targets at the tick-pathogen interface. In this regard, some reports in *R. microplus*, shown that RNAi silencing of some transcriptional factors of Toll, IMD, and Jak/STAT signaling pathways, lead to consider IMD pathway as the main regulator in the midgut and salivary glands during *A. marginale* infection [33]. The bacterial load also can be modulated by RNAi, as in *Amblyomma maculatum,* where the silencing by RNAi of the antioxidant selenoprotein thioredoxin reductase (TrxR) caused a reduction in the bacterial load of *Rickettsiaceae* in the midgut and salivary glands [38]. Additionally, by silencing of glutathione peroxidase (GPx) gene, the load of *B. burgdorferi* is reduced in *I. scapularis* saliva [39]. In other model systems, the crosstalk between RNAi and arthropods immune system defense has been explored. As in *Culex* mosquitoes, where Dcr-2, a central component of the siRNA pathway, recognizes West-Nile virus (WNV) dsRNA and activates a signaling cascade. This signaling stimulates Relish, increasing the expression of gene Vago (cytokine functional homolog) and activating the JAK-STAT pathway leading to an effective antiviral response [40].

These studies demonstrate that RNAi silencing of arthropod’s immune response genes is an approach that allows identifying vaccine candidates. Nevertheless, the application of this tool as tick control requires optimization of the delivery method of the dsRNA; besides the use of microinjection and soaking, probably the use of exosomes as a vector for dsRNA delivery could contribute to a successful strategy against vectors [41].

### 3.2. The Cellular Immune Response

The hemocytes are the surrounding cells in the hemolymph. In ticks, hemocytes are the main effectors of the cellular immune response to control pathogens by phagocytosis, encapsulation, and nodule formation (Figure 1) [42,43]. PAMPs on the microbial surface regulate these mechanisms. Phagocytosis is a primary cellular defense response to recognize invading pathogens, destroying them by endocytosis in phagolysosomes. This process might be regulated in ticks by a complement-like system composed of thioester-containing proteins, fibrinogen-related lectins, and convertase-like factors [6,42]. At the moment, there exist evidence that in *Ixodes ricinus,* C3-like molecules (IrC3-1, IrC3-2, IrC3-2), considered to be analogous molecules of the complement system, facilitate the phagocytosis of yeasts and *Borrelia* spirochaetes (Figure 1) [44].

On the other hand, encapsulation is a process that implies hemocytes accumulation around the parasite, foreign objects, or microbes, forming organized concentric layers. Specific lectins participate as opsonizing molecules that cause bacteria aggregation in the nodulation, but little is known about encapsulation and nodulation mechanisms. In this regard, both processes have been reported in *Dermacentor variabilis*. After inoculating this tick with *Escherichia coli,* the hemocytes form aggregates around the bacteria, a characteristic feature of nodules formation. Although the encapsulation process was evaluated in an artificial implant of Epon-Araldite inoculated under the tick cuticle, it remains to know whether encapsulation also occurs against microorganisms and if this process links with hemolymph coagulation and cellular response against some pathogens [45,46].

## 4. The Humoral Immune Response of Ticks

In ticks, this response consists of soluble effector molecules, called “humoral factors,” including AMPs, lysozymes, defensins, coagulation factors, proteases, and protease inhibitors (serpins, cystatins), enzymes involved in the oxidative burst and detoxification, as well as other recognition molecules (Figure 1) [42].

### The Tick Repertoire of Humoral Factors

The AMPs are considered ancient evolutionary systems of immune defenses based on humoral components widely distributed in nature [47]. The size of AMPs is usually small (below 10 kDa) and has cationic character. However, some anionic peptides have been reported, and they also differ in their amino acid sequence and mode of action (Figure 2) [48].

AMPs are expressed mainly in hemocytes, fat body, gut, ovaries, and salivary glands. They respond to either blood-feeding or microbial challenge, and their release to the hemolymph is essential in the immune system response [11,17,49]. The antibacterial mechanism of AMPs targets Gram-positive bacteria, yeast, and fungi [50]. Interestingly, ticks use some proteins from the blood meal as a source for the production of AMPs, as in hemoglobin-derived AMPs (hemocidines) produced by the proteolytic activity of proteinases present in the tick gut [51].

Lysozymes are another type of molecule with a broad microbicidal spectrum. These digestive enzymes hydrolyze N-acetyl-muramic bonds and N-acetyl-D-glucosamine residues of the bacterial peptidoglycan. In *D. variabilis,* lysozymes are more abundant in hemolymph than in other organs and synergize with some defensins, breaking the bacterial cell wall and accelerating some AMPs actions [42].

Several immune responses involved in the recognition and control of pathogens depend on the activity of specific proteases or peptidases. Serine proteases are key regulating molecules for several immune responses, including coagulation, AMPs synthesis, and protein degradation [52]. In addition, the activation of this pathway is controlled by three serine proteases: factor C, factor B, and pro-clotting enzymes, which are activated in the presence of lipopolysaccharide (LPS) and released to the hemolymph that in turn, cause the immobilization of binding pathogen [53]. Protease inhibitors control various proteolytic pathways that play an essential role in tick immunity [54]. Their functions include antimicrobial activity, egg production regulation, development, thrombin, trypsin, and elastase inhibition [55]. Interestingly, the pan protease inhibitors of alpha-macroglobulin type are in the tick hemolymph, where they protect the ticks against invading microbes’ proteases [56].

## 5. Reactive Oxygen Species and Oxidative Stress

The hemocytes can produce Reactive Oxygen Species (ROS) in response to pathogens, which are eliminated by oxidative burst probably mediated by protein kinase C [42]. For instance, in hemocytes of *R. microplus*, the production of superoxide (O_2_^−^) and hydrogen peroxide (H_2_O_2_) is observed in response to microbial challenge with *Micrococcus luteus* (Gram-positive bacteria), zymosan, and phorbol 12-myristate 13-acetate (PMA), conversely, some authors propose that LPS is not able to activate tick hemocytes to produce ROS in vitro conditions. On the other hand, the tick hemocytes produce ROS during phagocytosis similar to that found in invertebrates’ phagocytes [57].

Interestingly, the participation of ROS in cell signaling mechanisms has been suggested, including the formation of extracellular traps (ETs) activated in response to pathogens. Still, this mechanism is not described in ticks [58,59].

Ticks possess several molecules that participate in oxidative stress, including heat shock proteins (HSPs), ferritins, and vacuolar ATPases (V-ATPase), among others, that have a role in tick survival and vector competence [60]. For instance, HSP participates in tick immune response helping to decrease pathogen proliferation [61,62]. Additionally, ferritin has antioxidant activity by sequestration of iron from the pathogens, which decreases pathogen proliferation. Finally, the V-ATPases have an activity in phagosome acidification that leads to a decrease in pathogen proliferation [63,64,65].

## 6. Do Ticks Have Extracellular Trap Formation?

NETosis is a mechanism present in neutrophils of mammals activated in response to antigen and pathogens recognition. This activation results in the chromatin decondensation, the disintegration of the nuclear envelope, and rupture of the cytoplasmic membrane, with the consequent release of chromatin in the form of traps. Previously, the DNA that composes the chromatin associates with granules containing enzymes and antimicrobial peptides known as Neutrophil Extracellular Traps (NETs), which contribute to the elimination of the pathogens [66,67,68].

In neutrophils, NETosis can be stimulated in vivo by biological agents such as bacteria, protozoa, fungi, and some viruses, or in vitro by chemical agents such as calcium ionophores, PMA, glucose, activated platelets, ultraviolet light, lipopolysaccharides, and bacteria [69,70,71,72,73]. At the moment, in neutrophils, two types of NETosis have been described, “late suicidal NETosis” and “early vital NETosis” (Figure 3).

In late suicidal NETosis, traps release occurs in response to biological or chemical agents that bind to Toll-Like Receptors (TLRs) type 2, 4, or 8. This binding causes an increase in the concentration of intracellular calcium that activates NADPH oxidase and the Raf/MEK/ERK pathway, which culminates in ROS production. Then, ROS act as second messengers for expressing myeloperoxidase (MPO) and neutrophil elastase (NE). Thus, forming a complex that contributes to the chromatin decondensation by citrullination of histones. ROS contributes to increased phosphorylation and expression of MAP kinases, resulting in the nuclear envelope and plasmatic membrane degradation. Finally, DNA releases as extracellular traps neutralizing and killing pathogens in the networks by the antimicrobial peptides, cathepsin G, and lactoferrin in the primary granules (Figure 3a) [66,68,69,73,74,75,76].

In contrast, in early vital NETosis, the biological activators recognize the TLR2 and TLR4 receptors on the neutrophil surface without rupturing the nuclear envelope and cytoplasmic membrane. Then, the loss of the shape of the nucleus and the initial formation of vesicles containing the chromatin and the synthesis of primary granules occurs. Finally, the vesicles containing the chromatin and the primary granules fuse with the cytoplasmic membrane to release the DNA as extracellular traps and eliminate pathogens (Figure 3b) [76]. The eosinophils and mast cells also release ETs, and some other cells such as monocytes and macrophages release ETs but to a lesser extent when compared to neutrophils [77,78]. Specifically, in the case of eosinophils, they can release their mitochondrial DNA in a death-independent way [78].

Despite its importance, the mechanism of NETosis demands more studies in invertebrates (oysters, mussels, and shrimps), where is known as ETosis. In these organisms, the hemocytes release ETs, which are stimulated by biological inducers that include lipopolysaccharides, PAMPs, bacteria, and damage-associated molecular pattern molecule (DAMs), and chemical inducers such as PMA; although these stimuli do not always result in the activation of traps [79,80]. Additionally, the DNA decoration comprises globular proteins such as elastase, myeloperoxidase, and calprotectin; however, their presence in ETs from invertebrates has not been reported, only the Heat Shock Protein 27 (HSP27) and the C-type lysozyme in ETs from the earthworm *Eisenia andrei* and the kuruma shrimp, respectively [81,82]. The study of ETosis in invertebrates has proposed that some activators or effectors of the mechanism may have homologous proteins. In ticks, it has been suggested that the proteins involved in chromatin decondensation are the HSP27 and peroxinectin (PXN), which may function analogously to myeloperoxidase of neutrophils. PXN is a secreted opsonin that mediates cellular adhesion to pathogens, enhances the antimicrobial oxidative burst in hemocytes, and regulates granule exocytosis and encapsulation of invasive bacteria [83,84]. Another suggestion points out that a similar activity of the antimicrobial peptide Cathepsin G may be performed by ixodidin, a protein probably contained in the primary granules present in the NETs [68,81,85]. Although ETosis is a crucial mechanism in the defense against pathogens, most of its signaling pathways are unknown in ticks. So far, we know that ETosis is an evolutionarily conserved invertebrate defense mechanism that they have used for different metabolic and immunological processes [74]. In addition, ticks may use their AMPs for the granule formation of the ETs that eliminate the pathogen trapped. On the other hand, the presence of ROS may have a vital role in the activation of ETs in ticks, as described in NETosis, as in *A. marginale*, that manipulate redox metabolism in *R. microplus*, and therefore the production or activation of signaling pathways effectors that promote proliferation [86].

## 7. A Comparison of Known, Proposed, and Potential Molecules: Towards an Integrative Vision to Vector Control

In this integrative vision, we described some of the molecules known and proposed as vaccine candidates. Additionally, we suggest looking at the molecules that participate in the ticks’ immune system as potential candidates that could be blocked, interfered with, overexpressed, repressed, or any other activities that contribute to either controlling or preventing tick infestations in cattle, and therefore TBDs.

Salivary glands and midgut proteins are an excellent potential target for vaccine development since they are determinants of pathogen transmission. In contrast, only a few molecules related to the tick immune response have been studied. As in tick saliva and midgut, where some antigens have shown low effectiveness, Klouwens et al. [87] found DNA vaccines against antigens Salp15, tHRF, TSLPI, and Tix-5 induced low to moderate IgG responses without inducing protection. On the other hand, the tick midgut is considered essential for blood digestion. In this regard, the only ectoparasite vaccine commercially available (in Australia and Cuba) targets Bm86, a midgut protein of *R. microplus*, and interferes with feeding and subsequent egg production [88]. Nevertheless, the effectiveness diminishes when tested in different geographical regions, for reasons so far unclear.

Other alternatives focus on gene silencing by RNAi because it is a good technique for screening antigens for vaccine development. In this regard, knockdown of genes involved in tick development, survival, and reproduction, as the vitellogenin receptor (VgR), identified in several tick species, blocked *Babesia* spp. into developing oocytes inhibiting their vertical transmission [89]. Additionally, other RNAi studies explore the importance of iron-binding proteins called ferritins, FER1 (intracellular), and FER2 (secreted) in *I. ricinus* and *Haemaphysalis longicornis.* The knockdown by RNAi of both ferritins results in a reduced blood-feeding capacity, high mortality after blood engorgement, and reduced fecundity due to iron overload [90].

The molecules involved in tick humoral response also are considered potential vaccine candidates. The recombinant cysteine proteases inhibitor cystatin-2 (HlCyst2) from *H. longicornis* overexpressed in the midgut and hemocytes slightly affect *Babesia bovis* growth in vitro assays, but its role in tick infection has never been experimentally examined [91]. Another example relies on *Rhipicephalus appendiculatus* cystatin 2a (Racys2a), where Parizi et al. [92] showed that recombinant cystatin (rBmcys2c) reduced the number of fully engorged adult females by 11.5%. Their role in inhibiting cysteine proteases during tick feeding makes cystatins powerful targets in this challenging anti-tick vaccine development. Other alternatives are cathepsins, proteins that, once silenced in *H. longicornis* (cathepsin B), can reduce blood digestion and transmission of *Babesia* spp. [93]. Additionally, some ferritins proposed as an anti-tick vaccine antigen showed good potential due to their role in successful blood-feeding and reproduction. Recombinant ferritins, HlFER1, and HlFER2 of *H. longicornis* used in rabbits’ vaccination trials showed that both ferritins are highly immunogenic, inducing host antibody production [94].

After the infestation challenge, ticks obtained from rHlFER1-immunized rabbits reduced the number of eggs and the number of ticks with completely hatched eggs, while rHIFER2 immunization showed a reduction in ticks body weight [90,94].

Finally, the proposal of ETosis in ticks and its resemblance with NETosis opens the possibility to explore another mechanism to control TBDs. An interesting NETosis approach is the reported by Gabriel et al. [95], where Lipophosphoglycan (LPG) present on the surface of *Leishmania donovani* showed protective activity against the anti-parasitic effect of NETs since mutant promastigotes lacking LPG showed less survival percentage upon the action of NETs as compared with wild type promastigotes.

It is essential to highlight that ETs are protective against invading pathogens as an antimicrobial defense strategy; however, an excessive trap release can have detrimental effects [80]. In contrast, a combination with vitamin C (VitC) may provide a protective effect in sepsis by inhibiting the generation of excess NETs, as observed in mice lacking the ability to synthesize VitC in which excessive NETs formation caused tissue damage [96].

In ticks, multifunctional proteins are part of the mechanism of ETosis, as PXN, which may act as an analog of MPO, a key enzyme for NETosis. In addition, this protein is essential in the tick immunological process, including cell adhesion and opsonization. In this regard, PXN presents peroxidase activity associated with an efficient microbicidal attack system against invading microorganisms. PXN has an essential role in melanization in other arthropods, where melanin acts as a protective barrier to defense against pathogens [97,98].

In summary, many of the molecules that participate in the mentioned mechanisms could be attractive and potentially used to combat ticks and Tick-borne pathogens (TBPs).

## 8. Conclusions

The ticks hematophagous condition generates an essential interaction between TBP, the vector, and the host. Within this triad, the pathogen-vector interaction highlights the importance of the tick’s immune system as a modulator of pathogen establishment and replication, and probably also in its transmission. Currently, there are examples of some arthropods, where the immune system has been used as an alternative and/or strategy for the control of vectors and the pathogens they transmit.

In addition, it is essential to note that the immune system is also involved in different biological processes for the survival of ticks, such as detoxification and reproduction, and even vector capacity. In this regard, this work suggests the existence of ETosis, a mechanism never before described in ticks and that it could be an essential component in the pathogen-vector immunological interaction.

Finally, the elucidation of the molecular mechanisms involved in the immune response of ticks opens the possibility of identifying essential components (genes, proteins, effector molecules) that can lead to the discovery of targets against tick infestations and the development of strategies to control tick-borne diseases.

## Figures and Tables

**Figure 1 pathogens-10-01511-f001:**
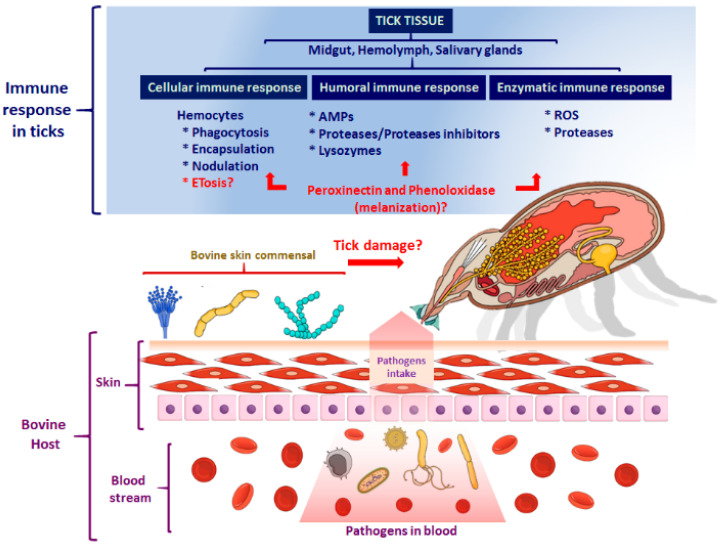
Immunological responses in a tick *R. microplus* during tick-bovine interactions. The immune response in ticks comprises cells, molecules, and proteins acting in combination to destroy pathogens. The tick also should afford the potential risk that would represent the commensals located in the skin of the bovine that, in turn, could be potentially infectious to the tick. Figure created in the Mind the Graph platform (www.mindthegraph.com, accessed on 9 September 2021).

**Figure 2 pathogens-10-01511-f002:**
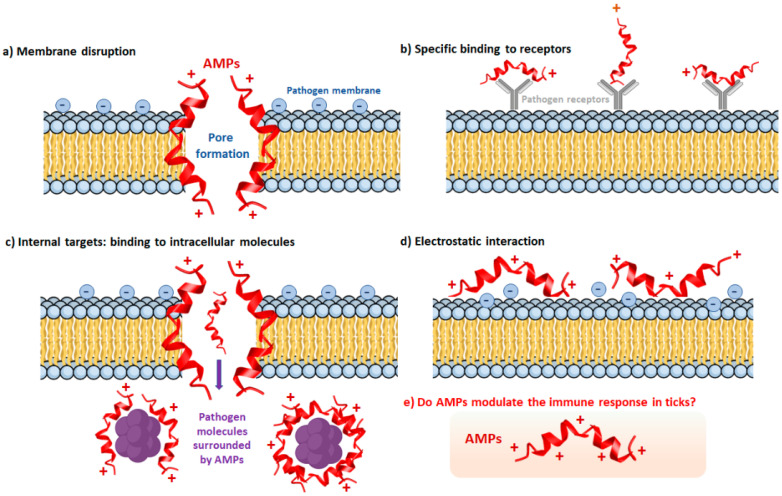
Mechanisms of action of the AMPs. The mechanisms of action of the AMPs vary depending on the interaction with the components of the cytoplasmic membrane. (**a**) Membrane disruption. The AMPs disrupt the integrity of the pathogen’s membrane by forming pores that eventually destabilize the structure and leak the internal content. (**b**) Specific binding to receptors. Some AMPs bind specifically to pathogen receptors, causing a steric hindrance and avoiding the correct binding to the ligands. (**c**) Internal targets: binding to intracellular molecules. The AMPs also aim at targets inside the pathogen, where they reach the intracellular molecules blocking their function. (**d**) Electrostatic interaction. The electrostatic interactions between the AMPs and the phospholipids on the pathogen’s surface change the membrane potential, resulting in deficient activity of the membrane components with the external environment. (**e**) Do AMPs modulate the immune response in ticks? The AMPs may be involved either in signaling pathways or activating processes of the immune response to pathogens. They may also be acting as main effectors in specific immunological mechanisms. Figure created in the Mind the Graph platform (www.mindthegraph.com, accessed on 9 September 2021).

**Figure 3 pathogens-10-01511-f003:**
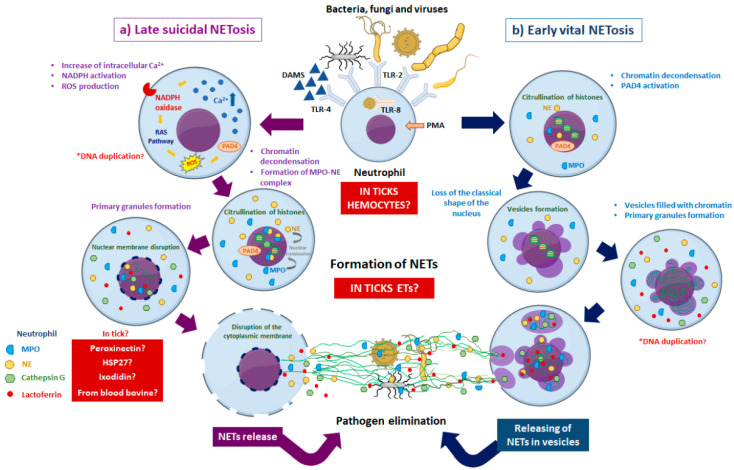
Main events in the mechanisms of late suicidal and early vital NETosis for the destruction of pathogens. (**a**) Late Suicidal NETosis. After recognizing biological and/or chemical activators, an increase in ROS production occurs that activates molecules related to the decondensation of chromatin, leading to the disruption of the cytoplasmic membrane and the formation of primary granules. The DNA and the content of the granules are released to the exterior as traps against the pathogens. (**b**) Early Vital NETosis. After recognition of biological and/or chemical activators occurs a ROS-independent activation, followed by DNA decondensation, citrullination of histones, and vesicles formation containing chromatin. These vesicles travel to the cytoplasmic membrane and release their content to the outside without damaging its nuclear envelope. The formation of ETs in the hemocytes of the tick may be occurring as a protection mechanism to cope with the presence of pathogens. The participation of peroxinectin, HSP27, and ixodidin has been suggested during the formation of these extracellular traps; however, in ticks, these immunological mechanism remains unclear. In the crab *Carcinus maenas* and the bivalve mollusk *Crassostrea gigas,* ROS production in ETs formation occurs only in an NADPH oxidase-dependent pathway. Until now, the information about NADPH oxidase independent pathway ETosis is unknown in invertebrates. Figure created in the Mind the Graph platform (www.mindthegraph.com, accessed on 9 September 2021).

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
