# Peer review of "Tick Immunobiology and Extracellular Traps: An Integrative Vision to Control of Vectors"

_pathogens, 2021, doi:10.3390/pathogens10111511_

Round 1

Reviewer 1 Report

This is a revised resubmission on a previously submitted review that aims to cover very broad topics relating to tick immunity and the development of vaccines to combat tick-borne pathogen transmission. The revisions included by the authors have improved the quality of this review, however there are still some outstanding issues. Suggestions for the authors are noted below.

Abstract: The last sentence does not make sense and needs to be restructured. When the authors refer to the ticks “natural enemies” are they meaning tick transmitted pathogens? Tick-transmitted pathogens do not negatively impact the tick itself, so are therefore not enemies.

Lines 97-100: run on sentence. Please split into at least two statements.

Line 119: Toll should be “Toll pathway”

Line 133: Immune deficiency (IMD) should be “Immune deficiency (IMD) pathway)

Line 152: Janus Kinase/Signal Transducer and Activator of Transcription (JAK/STAT) should be “Janus Kinase/Signal Transducer and Activator of Transcription (JAK/STAT) pathway”

Line 166: RNA interference (RNAi) should be “RNA interference (RNAi) pathway”

Line 139: DRED should be DREDD

Line 145: It is unclear what the “transcription factor basket” is, as the cited reference [28] does not mention a transcription factor basket.

Line 156: the wording in this sentence sounds like JAK/STAT is activated in bacteria. Consider changing the wording to “In ticks, this pathway is activated by bacteria…”.

Line 284: serin should be serine

Lines 444-448: The later half of the paragraph talks about NETosis as a contributor to pathology seen in COVID-19 patients and discusses use of DNAse as a therapeutic. How this fits with the theme of targeting NETosis in ticks for controlling tick borne disease is unclear.

Lines 449 – 453: Although this is interesting, it is unclear why the authors are discussing therapeutics to treat the pathological side effects of ETs when this section is described as “an integrated vision to vector control”.

Line 455: PXN has already been defined earlier

Line 462 – 465: “Therefore, we cannot discard the possibility that tick immune system targets may be recognized by host antibodies and then interfere with their immunological functions, increasing the susceptibility of ticks to natural enemies that are used in biological control [97].”

This sentence is uncomfortably similar to a sentence written in another previously published article with some of the same authors from this submission doi.org/10.1155/2020/8882031, “So, we cannot discard the possibility that peroxinectin may be recognized by host antibodies and, then, interfere with its immunological functions, increasing the susceptibility of ticks to external microorganisms that are used in biological control [37].”

Line 491: “TBP” needs to be defined before using the acronym.

Reviewer 2 Report

This review paper describes tick immune mechanisms that can be potentially targeted for the development of control strategies against tick infestation and tick-borne diseases. The subject is definitely relevant considering the current emerging importance of ticks and tick-borne diseases have for humans and livestock. Despite that, to me, the authors failed to make a clear connection between tick immune mechanisms and development of anti-tick strategies. Below are my general and minor comments.  

General comments:

- In general, the authors did not make a clear connection between tick immune mechanisms and potential targets for anti-tick-vaccines. For most of the sections in the manuscript, the reader is presented by a large body of knowledge of tick immunology, but it is really hard to connect the dots. For instance, how the components of tick immune signaling pathways can be explored for the development of control strategies (?). The authors should present and discuss the data considering that most of the pathway molecules are intracellular and consequently, not promptly available for immune responses of the vertebrate host.

- The same goes for ROS (and molecules involved in the oxidative stress). How can these molecules and enzymes associated with ROS and NO production be reasonable targets for the vertebrate immune effectors?

- The authors also failed to connect RNAi and development of anti-tick strategies/vaccines. To me, RNAi is a defense mechanism of ticks against pathogens, and it can also be utilized as an approach for gene functional analysis (including functional analysis in the tick model). However, it is hard to connect RNAi and anti-tick vaccines. Therefore, it would be interesting to see the author’s opinion on the subject.

- The use of “integrative vision” sounds as a jargon to me, and unless very well defined, it should be avoided.

Minor:

- Check abbreviations and the use of acronyms.

- Manuscript needs English revision.

- The word “however” is misused in several parts of the manuscript.

- Lines 301-304 Confusing and poor English.

- Line 39. A reference is missing after 14-19 billion US$/year.

- Lines 159-162. Confusing and poor English.

- Figures are good and illustrative; however, Fig 3 may be out of context since it is based on a mechanism described in neutrophils and still unclear in ticks, as mentioned by the authors.

- Lines 444-448. References to Covid-19 should be deleted.

- Lines 436-438. Confusing and poor English.

- Lines 466-467. Disconnected sentence.

- Lines 489-494. Sentences need English revision.

Reviewer 3 Report

Review of the paper 1401990 “ Tick immunobiology and Extracellular Traps: an integrative vision to control of vectors" by Aguilar-Diaz et al. for the Pathogens

I congratulate the Authors on their concept.

I enclose a few detailed suggestions which I hope will improve their manuscript.

With best regards

Your Reviewer

Abstract

Taking into consideration the circulation of tick-borne pathogens and life cycles of ticks the term “definitive hosts” shouldn’t be used in this context. Delete definitive leave just “hosts” or “human and animal hosts”

Line 25- please explain the abbreviation ETosis. When we introduce a new term in the text, we do not use the abbreviation itself.

Introduction

Lines 39-43: did you mean certain geographical areas? the only reference cited concerns data from India

Line 40: Please consider using the term prevalence instead of incidence.

Line 44 Please change ticks biting to tick bites

While mentioning direct consequences of tick feeding you missed toxicoses e.g. tick paralysis. Add missing information in the first sentence of the second paragraph.

  1. Physical barriers and tissues

Line 87: Change cement to cement precursors (please see Suppan J, Engel B, Marchetti-Deschmann M, Nürnberger S. Tick attachment cement - reviewing the mysteries of a biological skin plug system. Biol Rev Camb Philos Soc. 2018;93(2):1056-1076).

  1. The tick immunobiology

Line 139 Shouldn’t it be DREDD? (see Fogaça AC, Sousa G, Pavanelo DB, Esteves E, Martins LA, Urbanová V, Kopáček P, Daffre S. Tick Immune System: What Is Known, the Interconnections, the Gaps, and the Challenges. Front Immunol. 2021;12:628054)

None of the developmental stages of ticks have two pairs of legs. Three pairs of legs are present in larvae, while four pairs are in nymphs and the adults. Even simplified schemes should reflect the basic morphology correctly. I suggest adding one or two pairs of legs.

See figures in eg.  

Hovius JW, van Dam AP, Fikrig E. Tick-host-pathogen interactions in Lyme borreliosis. Trends Parasitol. 2007;23(9):434-8. (Figure 1)

Fogaça AC, Sousa G, Pavanelo DB, Esteves E, Martins LA, Urbanová V, Kopáček P, Daffre S. Tick Immune System: What Is Known, the Interconnections, the Gaps, and the Challenges. Front Immunol. 2021;12:628054 (Figure 1)

  1. Reactive Oxygen Species and Oxidative stress

Line 297: Superoxide symbol should be corrected

Line 306: change HSP to HSPs

Section 6 Mechanisms of ETosis in ticks

Lines 322 and 423: in vivo as well as in vitro should be written in italics throughout the text

Line 347: change TLR to TLRs

  1. A comparison of known, proposed, and potential….

Line 418: Change “blocked Babesia spp. transmission and inhibit their vertical transmission”

to “blocked Babesia spp. transmission  into oocytes and inhibit their vertical transmission”

Lines 428-430- it seems that there is a lack of reference cited.

  1. Conclusions

Please remove the first paragraph (lines 471-475) and delete the word   "soon" from the last one.

Line 491: Change TBP to TBPs

References

Line 696: incorrect citation of surnames in the publication by Mitchell et al. 2019

Mitchell RD 3rd, Sonenshine DE, Pérez de León AA. Vitellogenin Receptor as a Target for Tick Control: A Mini-Review. Front Physiol. 2019;10:618.

Round 2

Reviewer 1 Report

The authors made the requested changes, which has improved the manuscript.

Reviewer 2 Report

The authors have addressed most of my concerns.  One minor comment: I am not sure why the subtitle for section 6 has a question mark (?). If the subtitle is a question then, it should be properly state (e.g., "Do ETosis play a role in ticks?" or Do ticks have ETosis?"). Other than that, I have no further comments.

Author Response

This manuscript is a resubmission of an earlier submission. The following is a list of the peer review reports and author responses from that submission.

Round 1

Reviewer 1 Report

In this study, the authors present and discuss several tick immune mechanisms and molecules as potential targets for the development of anti-tick vaccines. The study is important by compiling published data that can be used for the design of novel strategies for tick control. Despite of being very informative, the study fails to clearly demonstrate a direct connection between tick immune mechanisms/molecules and development of anti-tick vaccines. My comments are below.   

Major comments

  • In general, the study fails to present a rationale of how the tick immune mechanisms and molecules can be used for the development of anti-tick vaccines. For instance, it is not clear to me how tick immune molecules present in the hemolymph can be targeted by the vertebrate immune system. Are the authors thinking/proposing that vertebrate immune cells and/or antibodies will block the tick immune molecules? If so, challenges associated with the potential success of this strategy need to be discussed.
  • The authors discuss potential targets for anti-tick vaccines. However, there is a controversial discussion that by elucidating a strong vertebrate immune response against tick saliva components may induce undesirable skin inflammation. The author may want to discuss this issue.
  • In several parts of the manuscript, the authors discuss experiments that used RNAi. My understanding is that RNAi is a gene silencing approach to study function. I would appreciate if the authors could present a rationale for connecting RNAi results and development of efficient vaccines.
  • The authors need to clarify and exemplify how tick hematocytes (or molecules produced by these cells) can be used in vaccine formulations.
  • It seems to me that nodulation, melanogenesis, coagulation can be potentially targeted as anti-tick vaccines; however, the authors failed to make a connection. For instance, how the cellular and humoral immune system of vertebrated can be efficiently induced against nodulation/melanogenesis/coagulation molecules to control tick infestation.    

Minor comments:

  • Lines 132-133 – Sentence needs revision for clarity.
  • Line 467 – It seems to me that reference 34 is misplaced, since the cited study does not address transmission of Babesia parasites.

Reviewer 2 Report

The topic of this review, New insights on the immune system of the tick reveal targets for vaccine design, is an interesting subject. The authors attempt to cover a very broad topic ranging from tick organs, saliva molecules, physical barriers, soluble molecules in the hemolymph, AMPs, cellular immunity and other enzymes, existing vaccine strategies and the possibility of ETosis in ticks. One intriguing aspect of this review is the speculation of ETosis in ticks and how this may impact tick immunity. This section goes into a good amount of detail and is an exciting take on tick immunity.

While the topic of vaccine design against tick immune molecules is interesting, the subsections covered in this review are too broad and do not go into enough depth. Several major topics within tick immunity are not discussed and the organization is confusing with text in some subsections that do not line up with major headings.  Suggestions for the authors are noted below.

Major comments:

  1. There is insufficient detail in subsections covering tick immunity.
  2. The organization of subtopics is not intuitive. For example, section 2 is titled “the immune system of ticks”, but instead discusses physical barriers such as the midgut, salivary glands and peritrophic membrane. Sections 3 and 4 describe cellular and humoral immunity, which should be under section 2.
  3. In Section 2.2.3. Fat body, only insects are discussed. This section should include some discussion about the tick fat body, citing primary research.
  4. The authors neglect to discuss major humoral immune pathways that trigger production of AMPs such as the Toll pathway and the IMD pathway.
  5. Authors mention that ticks transmit viruses as well as bacteria and parasites, but do not discuss antiviral immune mechanisms.
  6. Under the subsection titled “Vaccine designing based on molecules of the immune system in ticks” the authors discuss vaccines targeting physiological aspects of the tick such as salivary molecules, midgut molecules and enzymes needed for blood digestion, which are not components of the immune system.
  7. There is a good amount of detail on the intriguing topic of NETosis, but is not balanced with the amount of detail in the arthropod immunity sections.
  8. A vaccine targeting ETosis is an interesting idea. Readers would benefit from extended discussion on this topic.
  9. Citing primary research papers in place of reviews is recommended.

Minor comments:

  1. Some paragraphs only have 1 sentence. Combining these sentences with other content is suggested and will improve readability.
  2. First use of the acronym TBP in the conclusions paragraph is not spelled out (tick-borne pathogens).